# COL7A1 Expression Improves Prognosis Prediction for Patients with Clear Cell Renal Cell Carcinoma Atop of Stage

**DOI:** 10.3390/cancers15102701

**Published:** 2023-05-10

**Authors:** Dzenis Koca, Irinka Séraudie, Rémy Jardillier, Claude Cochet, Odile Filhol, Laurent Guyon

**Affiliations:** Interdisciplinary Research Institute of Grenoble, IRIG-Biosanté, University Grenoble Alpes, CEA, INSERM, UMR 1292, F-38000 Grenoble, France

**Keywords:** ccRCC, patient’s prognosis, COL7A1, biomarker, prediction, survival

## Abstract

**Simple Summary:**

Clear cell renal cell carcinoma (ccRCC) is the most common subtype of kidney cancer. Although it ranks as the seventh most diagnosed type of cancer, ccRCC is increasingly prevalent in the Western world. While most cases of ccRCC are diagnosed in the early stages, the cancer can often progress, spread to distant organs, and recur after treatment. Assessing the survival of ccRCC patients, and thus the aggressiveness of the cancer, can help clinicians make more informed decisions on patient management. We employed a systematic approach to find new biomarkers that can improve patient survival prediction. Our research revealed that increased expression of the COL7A1 gene is correlated with an aggressive form of the cancer and improve patient survival assessment, complementing the clinical characteristics that are already in use for survival prediction. Furthermore, our findings indicate that COL7A1 is a robust biomarker, that can be applied to patients of different origins and who undergo different treatment procedures. Interestingly, preliminary experiments led to the hypothesis that COL7A1 may play a role in cancer progression.

**Abstract:**

Clear-cell renal cell carcinoma (ccRCC) accounts for 75% of kidney cancers. Due to the high recurrence rate and treatment options that come with high costs and potential side effects, a correct prognosis of patient survival is essential for the successful and effective treatment of patients. Novel biomarkers could play an important role in the assessment of the overall survival of patients. COL7A1 encodes for collagen type VII, a constituent of the basal membrane. COL7A1 is associated with survival in many cancers; however, the prognostic value of COL7A1 expression as a standalone biomarker in ccRCC has not been investigated. With five publicly available independent cohorts, we used Kaplan–Meier curves and the Cox proportional hazards model to investigate the prognostic value of COL7A1, as well as gene set enrichment analysis to investigate genes co-expressed with COL7A1. COL7A1 expression stratifies patients in terms of aggressiveness, where the 5-year survival probability of each of the four groups was 72.4%, 59.1%, 34.15%, and 8.6% in order of increasing expression. Additionally, COL7A1 expression was successfully used to further divide patients of each stage and histological grade into groups of high and low risk. Similar results were obtained in independent cohorts. *In vitro* knockdown of COL7A1 expression significantly affected ccRCC cells’ ability to migrate, leading to the hypothesis that COL7A1 may have a role in cancer aggressiveness. To conclude, we identified COL7A1 as a new prognosis marker that can stratify ccRCC patients.

## 1. Introduction

In the year 2020, around the world, there were an estimated 431,300 new cases of kidney cancer, accounting for 2.2% of all cancer cases, and nearly 179,300 patients died of this disease [1]. Clear-cell renal cell carcinoma (ccRCC) is a distinct histologic subtype of kidney cancer that accounts for approximately 75% of kidney cancer cases [2]. The ccRCC subtype is histologically characterized by having transparent (clear) round-shaped cells. Multiple cancer driver events were identified through the molecular characterization of ccRCC. Some of the most common driver events include mutations and methylation differences for VHL, PBRM1, BAP1, and SETD2 genes, as well as changes in chromosomal structure such as loss of 3p and gain of 5q chromosomes [3]. Over the years, the prognosis of ccRCC patients has improved, although its incidence is indeed increasing [4]. According to the 2019 data, around 17% of newly diagnosed patients presented with distant metastases, and the 5-year survival probability of these patients is less than 12% [4]. Early ccRCC cases are asymptomatic, and they are often discovered accidentally during standard medical checkups and CT or ultrasound of the abdomen. Partial or radical nephrectomy remains a primary treatment for localized (stages I–II) tumors, with options of active surveillance (stage I patients) or applications of ablative techniques (localized T1a tumors) [5,6,7]. Still, reoccurrence occurs in around 30% of patients that undergo surgery, and this includes 10–25% of patients with localized disease [8]. For advanced ccRCC cases (stages III–IV), surgery remains the primary treatment (when the tumor is accessible), although adjuvant therapy is recommended [5]. Novel therapeutical options, such as targeted therapies and immunotherapies, often come with side effects and high costs and are often prescribed to patients with intermediate to poor prognoses [5,9]. Nevertheless, there is an urgent need for biomarker-driven studies with the overarching goal of incorporating newly proven biomarkers into innovative trial designs and giving patients a more accurate and individualized prognosis.

The extracellular matrix (ECM) plays an important role in many cancers. ECM can serve both as a mechanical support/scaffold for tumor growth and cell proliferation and migration as well as a modulator of biochemical cues by promoting epithelial-mesenchymal transition, sustaining self-renewal, inducing metabolic reprogramming, and accumulating and delivering various growth factors [10]. Collagen proteins are major structural and functional components of the ECM. Type VII collagen is a protein that belongs to the family of collagens and is encoded by the Collagen Type VII Alpha 1 Chain (COL7A1) gene. The collagen 7 fibril is formed by intertwining three identical collagen chains, and it primarily functions as an anchoring fibril between the basal membrane and proximal cells of stratified squamous epithelia [11]. Germline mutations in COL7A1 are one of the main causes of dystrophic epidermolysis bullosa, which in turn can increase the chance of squamous cell carcinoma [12]. Increased expression of COL7A1 was also detected in highly metastatic cancer cell lines as well as in prostate cancer-initiating cell spheroids [13,14]. COL7A1 expression has already been investigated as a potential prognostic biomarker in patients with lung squamous cell carcinoma [15], esophageal squamous cell carcinoma [16], gastric cancer [17], and pancreatic cancer [18].

Assessment of patient survival could improve decision-making in a clinical environment. Additionally, certain treatment options are allowed to be used only in cases of poor prognosis. Clinical and biochemical markers, such as stage and grade of tumor, are often used in the assessment of ccRCC prognosis [19]. Recently, we have shown that, for most cancers, survival prediction can be improved by incorporating clinical and transcriptomic data [20]. Additionally, there is an increasing interest in how ECM-related genes could be used to assess the survival of ccRCC patients [21,22,23,24]. To the best of our knowledge, COL7A1 expression in ccRCC was not investigated as an independent prognostic biomarker in this disease, and this is the first ccRCC study using five independent public datasets.

The present paper is organized as follows: Firstly, we performed a screening for potential biomarker genes in ccRCC, followed by an in-depth investigation of the top results. Secondly, we investigated the possible biological implications of COL7A1 through the assessment of co-expressed genes. Finally, we perform *in vitro* experiments to investigate the impact of COL7A1 expression on the proliferation and migration of ccRCC cell lines.

## 2. Materials and Methods

### 2.1. Summary of Methods

To facilitate an overview, we summarize here the methods that we further detail in the following paragraphs. We investigated the prognostic potential of COL7A1 expression in ccRCC tumors using five different cohorts, summarized in Table 1. We used univariable and multivariable Cox models, together with stage, grade, and age, to assess its prognostic value. We performed gene set enrichment analysis (GSEA) to investigate biological pathways affected in situations in which COL7A1 is overexpressed. Finally, we knocked down COL7A1 expression in an aggressive cell line and investigated the effect on proliferation and migration.

### 2.2. Datasets and Processing

The cancer genome atlas (TCGA) is a cancer genomics program that has characterized over 20,000 tumoral and peritumoral tissue samples from 33 different cancers, including ccRCC (with the acronym KIRC). We obtained the transcriptome profiling data of primary tumor tissue and the corresponding clinical information of the TCGA:KIRC cohort using the “TCGAbiolinks” R package (v2.62) [25]. As per a recent suggestion [26], we downloaded harmonized transcriptomic data, which was already transcripts per million (TPM) normalized. To decrease computational time, we retained only the protein-coding genes with total expression across samples greater than 10 TPM. Next, we applied the log2(TPM + 1) transformation to the data. To deal with sample multiplicity in GDC data (more than one sample per patient), we only analyzed samples that could be found in GDC “FireBrowse” (http://firebrowse.org/, accessed on 2 November 2022).

To validate the results on independent cohorts, we have obtained three additional datasets. We obtained the transcriptome profiling data for the E-MTAB-1980 dataset from Array Express, while we obtained the clinical data from supplementary data in the original publication [27]. We obtained the GSE167093 dataset from Genome Expression Omnibus (GEO) [28] and the “Braun *et al.*” and JAVELIN renal 101 datasets from the supplementary data of corresponding publications [29,30]. The original authors have already normalized all datasets of independent cohorts, and details about normalization are available in their respective publications. An overview of the clinicopathological characteristics of patients in datasets for which clinical data were publicly available is given in Table 1.

### 2.3. Univariable and Multivariable Cox Model

We applied two thresholds on gene expression: (a) filtering out genes that are not expressed in at least half of patients (median expression = 0, in log2(TPM + 1)), and (b) filtering out genes that are not variable enough to be precisely measured in a clinical environment (quantile 80–quantile 20 = 0, in log2(TPM + 1)). After the filtering process, we used each remaining gene to train a univariable Cox model. For this purpose, we used the R “survival” package (v3.3.1) [31,32] and we used the obtained *p*-values to judge the goodness of fit of the Cox model.

Next, we performed a univariable Cox model on clinical covariates:the age of a patient, gender, clinical stage, and histological grade, as well as COL7A1 expression. We then used all the mentioned covariates to learn a multivariable Cox model and compared changes in the *p*-value of the COL7A1 covariate between the univariate and multivariate Cox models. 

We also conducted an analysis of the deviance of the multivariable Cox model to determine the added value of COL7A1 expression compared to the multivariable Cox model based solely on the mentioned clinical covariates. The analysis of deviance table presents the improvement in a predictive power model as each of the covariates is added to the model, while the *p*-value shows whether the improvement was significant.

### 2.4. Kaplan–Meier Curves

We divided patients into groups of high and low COL7A1 expression using a cut-point determined by the “surv_cutpoint” function of the “survminer” package (v0.4.9) [33]. We applied the same process to the other three independent datasets. The division of patients according to COL7A1 expression into four groups was performed by dividing the outlier-free range of expression into five equal sections using the “cut” function. We computed and drew all Kaplan–Meier curves using the packages “survival” and “survminer”.

### 2.5. GSEA

To better understand the biological implications of COL7A1’s aberrant expression, we investigated genes that are co-expressed with COL7A1. Gene set enrichment analysis (GSEA) allows us to investigate how the expression of a set of genes that belong to a common biological pathway varies in expression together with COL7A1 [34]. We performed GSEA using the “ClusterProfiler” (v4.6.0) package [35]. For this purpose, we investigated co-expression through Pearson’s correlation between COL7A1 expression and the expression of other genes available in each investigated dataset. Gene sets (pathways) on which we have performed GSEA were obtained from the Molecular Signature database (MSigDB, v2022.1), with a focus on the Hallmark dataset [34,36].

### 2.6. Statistical Analysis and Data Manipulation

All of the aforementioned analyses were performed using the R software (v4.2.1) [37]. The set of packages contained in the “tidyverse” (v1.3.2) library (dplyr, ggplot, purr…) were used to manipulate data and create graphs [38].

### 2.7. Cell Culture

RCC cell lines 786-O, ACHN, and Caki-1 were obtained from ATCC (CRL-1932, CRL1611, and HTB-46, respectively). The cell lines were grown in 10 cm diameter plates in a humidified incubator (37 °C, 5% CO_2_) with RPMI 1640 medium (Gibco) containing 10% fetal bovine calf serum, penicillin (100 U/mL), and streptomycin (100 µg/mL).

### 2.8. Generation of shCOL7A1 786-O Cells

Four Lenti-pLKO.1-puro shRNA vectors that specifically targeted COL7A1 mRNA sequences were chemically synthesized (Sigma Aldrich, St. Louis, MO, YSA) and included in viral particles: (#1: 5′-CCGGCCCTTGAGAGGTGACATATTCCTCGAGGAATATGTCACCTCTCAAGGGTTTTTG-3′, #2: 5′-CCGGGCTCGCACTGACGCTTCTGTTCTCGAGAACAGAAGCGTCAGTGCGAGCTTTTTG-3′, #3: 5′-CCGGGAGCCAGTGGATTTCGGATTACTCGAGTAATCCGAAATCCACTGGCTCTTTTTG-3′, #5: 5′-CCGGGCTCGCACTGACGCTTCTGTTCTCGAGAACAGAAGCGTCAGTGCGAGCTTTTT-3′). 786-O cells were plated into 6-well plates in 2 mL of serum-supplemented RPMI 1640 medium. The day after, adherent cells were infected with COL7A1 virus (Sigma Aldrich) (1–5 MOI (multiplicity of infection)) diluted in 1 mL of serum-supplemented medium containing 8 µg/mL of polybrene (Sigma Aldrich). After 4 h, 1 mL of medium was added to each well, and transduction was maintained for 16 h before changing the medium. For stable transduction, puromycin selection started 36 h post-infection (at a concentration of 2 µg/mL) and was maintained for 1 month.

### 2.9. Western Blot

Proteins were extracted from confluent-plated cells in Laemmli buffer. Samples were heated for 5 min at 100 °C. They were separated on a NuPAGE 4–12% Bis Tris gel (BioRad, Hercules, CA, USA) at 150 V for 1 h 15 and electro-transferred to polyvinylidene difluoride (PVDF) membranes (BioRad) for 1 h at 100 V. Membranes were saturated for 1 h at room temperature in 3% BSA in TBS-Tween20 0.05% and then incubated overnight at 4 °C with the appropriate primary antibody diluted in the same saturation buffer. This was followed by incubation with horseradish peroxidase (HRP)-conjugated secondary antibodies and detection by enhanced chemiluminescence. Anti-GAPDH was used as a protein loading control. Antibodies used: COL7A1 (Santa Cruz Technologies #sc33710; 1/1000^e^, Goa, India) and GAPDH (Life technologies #AM4300; 1/40,000^e^, Carlsbad, CA, USA)

### 2.10. RT-qPCR

RNA extraction was performed from confluently plated cells. Trizol was added to the cell plates. After the addition of chloroform, the extraction was centrifuged, the resulting upper phase, which is RNA, was precipitated with isopropanol, and the pellet was washed twice with ethanol. RNA was recovered in RNAse-free water and dosed using Nanodrop.

Reverse transcription was achieved using the iScript cDNA synthesis kit (BioRad). Briefly, 1 µg of RNA was mixed with the 5x iScript Reaction Mix, reverse transcriptase, and water. Tubes were then incubated in a thermal cycler following this cycle: 5 min at 25 °C, 20 min at 46 °C and 1 min at 95 °C.

Real-time quantitative PCR (qPCR) was achieved thanks to the kit Prome GoTaq qPCR Master Mix (BioRad), according to manufacturer instructions. Briefly, pre-diluted (1/10) cDNA was added to the PCR plate as well as the reaction mix containing Taq polymerase and primers (10 µM) and completed with water up to a final volume of 10 µL. qPCR was run using the Biorad C1000 Thermal Cycler following this cycle: 95 °C for 2 min, 95 °C for 15 sec, and 60 °C for 45 sec, repeating those two steps 39 times and finishing with 95 °C for 10 sec and 65 °C for 5 sec. Primer sequences used are: #COL7A1_Fwd:5′-GGTGTTCCTACCACATGCCA-3′; #COL7A1_Rev:5′-GGAGGGCCGATGACTGTAAG-3′; #GAPDH_Fwd:5′-CCCATGTTCGTCATGGGTGT-3′; #GAPDH_Rev:5′-TGGTCATGAGTCCTTCCACGATA-3′; #RPL13_Fwd:5′-TTAATTCCTCATGCGTTGCCTGCC-3′; #RPL13_Rev:5′-TTCCTTGCTCCCAGCTTCCTATGT-3′. Relative quantification was performed using the comparative threshold (CT) method after determining the CT values for the reference and target genes COL7A1 in each sample set according to the E−∆∆Ct method. Changes in mRNA expression level were calculated after normalization to RPL13 and GAPDH mRNA.

### 2.11. Proliferation and Migration Assay

Proliferation and migration were assessed using an Incucyte ZOOM (Sartorius, Göttingen, Germany) video microscope. For proliferation, cells were plated in a 96-well plate at a density of 5000 cells per well, and cell confluence was measured by taking pictures every 2 h for 72 h. The cell number was also quantified by counting the cells at T-0 h and T-48 h in four different images for each condition (between 200 and 900 cells per image). The migration assay was performed by plating 30,000 cells per well in a 96-well plate. After overnight cell adhesion, a wound was made using the wound maker (Sartorius). Wound confluence over time was measured by taking pictures every 2 h during 24 h. An analysis of confluence was performed thanks to the Incucyte zoom software.

## 3. Results

### 3.1. COL7A1 Expression Is Prognostic of Overall Survival

With the aim of discovering novel prognostic biomarkers for ccRCC patients, we have applied a systematic approach to identify genes whose expression levels in tumors are related to overall patient survival. After the filtering processes that we described in the Material and Methods section, around 17,000 genes remained. We learned a univariable Cox model on all genes that passed filtering, and the resulting table was ordered according to the model’s *p*-value. The top ten genes with the lowest *p*-value are presented in Table 2, with COL7A1 being the one whose expression level is the most significantly linked to survival. To confirm the correlation of COL7A1 expression with survival, we trained a univariable Cox proportional hazards model on the independent E-MTAB-1980 dataset. The resulting model was significant, with a *p*-value of 0.0023. To visualize the prognostic capabilities of COL7A1, we have divided TCGA:KIRC patients into groups of high and low expression (Figure 1a). The resulting Kaplan–Meier curve shows that patients with high COL7A1 expression in tumors have a significantly worse prognosis (log-likelihood *p*-value < 0.0001) exhibiting a median survival of 4.3 years (3.3–5.25, 95% confidence interval (CI)) with a 5-year survival probability of 43.7% (34.75–52.5%). For patients with low COL7A1 expression, it was not possible to determine median survival as the survival curve does not cross the 50% line; however, this group had a 5-year survival probability of 73.1% (67.8–78.9%).

While the number of patients in the E-MTAB-1980 dataset is much lower compared to the TCGA:KIRC dataset, and patients are predominantly in the early stages, we can still see a similar stratification of patients according to risk (Figure 1b). Patients in the high COL7A1 expression group have a significantly worse prognosis than those in the low expression group (*p*-value = 0.0014), confirming the results obtained on the TCGA:KIRC dataset. The 5-year survival probability of the high expression group was 58.3% (39.8–85.5%), in contrast to 85.2% (77.4–93.7%) in the low expression group.

To further investigate the prognostic power of COL7A1, we divided the range of COL7A1 expression in the TCGA:KIRC dataset into four equal levels in terms of expression range. The lowest level of expression is almost identical to the low expression group in Figure 1a. The other three levels of expression showed a gradually increased risk and shorter survival time for patients in relation to COL7A1 expression levels (Figure 1c,d). The 5-year survival probability of each group from the lowest to the highest COL7A1 expression is 72.4%, 59.1%, 34.15%, and 8.6%, respectively.

### 3.2. COL7A1 Expression Can Predict Patient Survival Atop of Clinical Characteristics

Having investigated the predictive power of the standalone model based solely on COL7A1 expression, we opted to investigate how COL7A1 expression could be applied in real clinical settings. First, we investigated COL7A1 expression patterns across samples of different stages and histological grades. When compared to the overall expression of protein-coding genes in tumor samples, COL7A1 appears to be, in general, a weakly expressed gene (Appendix A), with a median expression of 0.8 (0.74–0.856), while the median expression of all protein-coding genes is 3.058 (3.055–3.06).

Additionally, when comparing the COL7A1 tumor expression across patients with different clinicopathological characteristics, we found that COL7A1 tends to be more expressed in patients of higher stage and grade (Supplementary Appendix A). Following this finding, we investigated whether COL7A1 stratification performances remain within these clinical characteristics (stage and grade). Indeed, we observed a significant stratification between patients with low and high COL7A1 expression in all stages (Figure 2). Stratification was most prominent in patients with stage II and IV tumors, with a *p*-value of 0.00099 and 0.00025, respectively. The 5-year survival probability for stage IV patients was 39.6% (95% CI 26.5–59.2%) for the low-expression strata and 11.6% (95% CI 5.1–26.5%) for the high-expression strata. The 5-year survival probability for stage I patients in low-expression (resp. high-expression) strata was 82.4% (resp. 69.3%), for stage II patients 87.3% (resp. 58.9%), and for stage III patients 63.5% (resp. 38.6%). We obtained similar results in two independent cohorts (Appendix A). In addition to improving prediction atop tumor stage, COL7A1 expression was able to improve survival prediction atop tumor grade as well (Appendix A). The most prominent effect (*p*-value of 0.00018) was observed in patients with grade 3 tumors, where the 5-year survival probability of low-expression (resp. high-expression) strata was 42.7% (resp. 13.6%). Although significant, the smallest separation was observed for grade 1 and 2 tumors, which already have a good outcome.

Additionally, we investigated the progression free survival of patients who underwent new treatment options (Appendix A). Significant separation of curves was obtained for both treatment options, with the best separation in Sunitinib-treated patients of the JAVELIN renal 101 cohort.

### 3.3. COL7A1 Expression Can Improve Multivariable Cox Model

Using the univariable Cox model, we investigated how survival prediction based on individual clinical variables compares to prediction based on COL7A1 (Table 3). Interestingly, COL7A1 expression showed better prediction than tumor grade or the age of patients. The only clinical variable that shows better prediction than COL7A1 expression is the tumor stage, especially for stage IV tumors. In a multivariable Cox model, COL7A1 keeps a particularly low *p*-value (*p* = 6.9 × 10^−9^), meaning that COL7A1 expression retains predictive power even when clinical characteristics are included in the model. To investigate what the added value of COL7A1 expression to the multivariable Cox model is based on clinical characteristics alone, we have computed an analysis of deviance based on the sequential partial log-likelihood. When variables were added to the model in the order they appear in the table, the COL7A1 expression showed significant improvement with a *p*-value of 9.6 × 10^−8^ and an increase in the log-likelihood of 28.45.

### 3.4. COL7A1 Expression Is Correlated with Genes Involved in Cell Division, Inflammatory Response, and Epithelial to Mesenchymal Transition and Anti-Correlated with Metabolism

To infer possible biological mechanisms, we have investigated how COL7A1 expression correlates with the expression of other protein-coding genes. As illustrated in Figure 3a, a GSEA analysis to identify the biological functions of correlated genes showed strong enrichment in cell proliferation pathways, inflammation/immune response, and epithelial to mesenchymal transition pathway. Moreover, Appendix A shows that proliferation-related genes are more expressed in tumors that also express high levels of COL7A1. On the other side, the GSEA analysis showed a strong downregulation in multiple metabolic pathways, with the strongest decrease being mitochondrial oxidative phosphorylation in patients with high tumor expression of COL7A1 (Appendix A). Additionally, all 12 mitochondrial genes expressed in the TCGA dataset and Braun *et al.* showed a clear anti-correlation pattern with COL7A1 (Figure 3b,d and Appendix A). Since the E-TMAB-1980 and GSE167093 datasets originate from expression arrays, they do not contain mitochondrial genes. To investigate the expression of genes for which the encoded protein is a part of mitochondria, we performed GSEA analysis on the cellular components (CC) of the Gene Ontology database. Most of the genes were found to be anti-correlated with COL7A1 (Figure 3c,e).

### 3.5. COL7A1 Knocked-Down Cells Show Decreased Migration Rate

As the analysis performed above shows correlation, we explored if the link between the high expression of COL7A1 and the low survival of patients could be causal. To validate the transcriptomic analysis, we performed *in vitro* experiments to investigate the impact of COL7A1 expression on the proliferation and migration of several RCC cell lines. First, RT-qPCR and Western blot analysis revealed differential COL7A1 expression in 786-O, ACHN and Caki-1 RCC cell lines, showing that the protein is abundantly expressed in the metastatic 786-O cells (Figure 4a,b,b′). Therefore, we generated shCOL7A1 786-O cells in which we knocked down COL7A1 expression by RNA interference using four different shRNAs targeting this gene. Cells infected with shRNA1 or shRNA5 COL7A1 showed a decrease in COL7A1 expression of 50 and 70%, respectively (Figure 4c,c′). Therefore, we chose these cells to further characterize them by cell proliferation and migration assays (Figure 4d,e). After 48 h of culture, control 786-O cells were almost confluent, whereas 10 and 20%, respectively, of the well surface were not covered shRNA1 or shRNA5 786-O cells, suggesting that proliferation was inhibited by 10 and 20%, respectively (Figure 4d and Appendix A). However, we noticed that shRNA1 and shRNA5 786-O cells were larger (more spread out) than shCTRL 786-O cells. Therefore, we quantified the cell number in each knockdown condition both at t-0 h and t-48 h to calculate their cell doubling time, which is directly correlated to proliferation (Figure 4d). This assay showed that COL7A1 knockdown in 786-O cells does not affect their cell proliferation. In contrast, cell migration of both shRNA cell lines assessed by a wound healing assay was significantly impaired (Figure 4e and Appendix A). 

## 4. Discussion

With the advancement of OMIC technologies and methods in machine learning, there has been huge progress in novel biomarker discovery in ccRCC [39]. In the present work, we have investigated the use of COL7A1 expression as a potential prognostic biomarker in ccRCC. COL7A1 is generally weakly expressed in tumor tissue (for around 70% of the patients), and high expression of this gene is a sign of poor prognosis. We have found this finding in independent cohorts from different ethnic groups, emphasizing its robustness. Additionally, COL7A1 expression is inversely proportional to overall survival, meaning that the higher the expression of COL7A1, the poorer the patient’s outcome. Consequently, COL7A1 shows potential to be used either as a binary prognostic factor (low or high expression) or as a continuous prognostic factor. Future predictive models may be improved by the incorporation of new biomarkers. By stratifying the Kaplan–Meier curve according to the stage and grade of tumors, we showed that COL7A1 can improve the prediction of patient survival based on these two variables. This feature is particularly important for patients with stage II or III cancers due to their high uncertainty of outcome. The TCGA and E-MTAB-1980 cohorts were completed before the use of recent treatment options, including mTOR and immune checkpoint inhibitors, which are given to patients with advanced stages. The prognostic potential of COL7A1 remains with these new therapeutic options, as we demonstrated with the Braun *et al*. and JAVELIN renal 101 cohorts.

Results of univariable and multivariable Cox models additionally support the claim that COL7A1 expression can improve prediction based on clinical variables only. Although our results on clinical variables are only comparable to previously published results [40], we want to comment on the hazard ratios (HR) of age and grade: (a) The HR of the age covariate is relatively low (1.03, Table 3), while the corresponding *p*-value is 2.5 × 10^−6^. The small 95% CI (1.018–1.045) explains this apparent contradiction, and the increased risk of 1.03 corresponds to a one-year difference, while the ages of the patients in the TCGA cohort span from 29 to 90 years old. (b) Grade 1 patients represent less than 3% of the TCGA cohort, so we decided to pool them with grade 2 patients. The HR of grades 3 and 4 thus corresponds mainly to a ratio with grade 2 patients. Additionally, histologic grade and stage are categorical variables, so to compare our results with studies treating them as continuous, one needs to divide by the corresponding grade/stage.

The GSEA analysis showed that high COL7A1 expression could be a sign of multiple other dysregulations in renal tumors. In particular, COL7A1 expression tends to be correlated with genes belonging to pathways that play key roles in cell proliferation, resulting in positive enrichment of “E2F targets”, “MYC targets”, “G2M checkpoint”, and “mitotic spindle” pathways. Alterations in these signaling pathways were observed in renal carcinoma patients [41,42,43,44], where they may lead to uncontrolled cancer-cell proliferation, promoting tumor progression and aggressiveness. The GSEA analysis shows strong positive enrichment in the “epithelial to mesenchymal transition” pathway, a signature of dedifferentiation of normal epithelial renal cells, as previously reported in a study comparing ccRCC cells to tumor-adjacent kidney tissue [45].

Our *in vitro* assays show that, unlike proliferation, migration of 786-O cells is reduced upon COL7A1 knockdown. This observation raises the hypothesis that high COL7A1 expression may have a causal effect on cell migration, facilitating invasion and metastasis, thus affecting patient survival. Such a hypothesis deserves a dedicated research project to be confirmed. In particular, we acknowledge that this *in vitro* work was conducted in 2D, which may be different from 3D conditions. Moreover, to relate prognosis prediction to COL7A1 expression, survival analysis of mouse xenografted with 786-O cells in which COL7A1 expression has been altered will be worth trying. If the role of COL7A1 in ccRCC aggressiveness is confirmed, it may be an interesting therapeutic target, as it is for type I collagen in some cancer types [46]. In addition, the pathways enriched in genes whose expression is correlated with that of COL7A1 would deserve specific attention.

We also observe anti-correlation between COL7A1 and mitochondrial genes, as well as genes involved in various metabolic pathways. Deregulations in multiple metabolic pathways as well as impairments in mitochondrial bioenergetics and oxidative phosphorylation are well known hallmarks of ccRCC [47]. Interestingly, a strikingly reduced mitochondrial respiratory capacity has been previously observed in primary human ccRCC cells [48]. In addition, the Warburg effect is known to be prominent in ccRCC and is known to play a role in epigenetic changes in cancer cells [49].

### Limitations

COL7A1 is rarely expressed in the tumors of the majority of patients. Thus, identifying patients with moderate risk would require sensitive mRNA measurement techniques, which are currently not widely available in clinics. For the same reason of low expression, this gene does not show up in single-cell RNA-seq data. Therefore, we were not able to pinpoint the exact cell type that expresses COL7A1 from publicly available datasets.

## 5. Conclusions

In the present work, we have shown that COL7A1 expression can stratify patients in terms of prognosis in independent cohorts, including recent ones dealing with up-to-date treatments. The prognostic capacity is higher than the grade and can further identify patients at risk within a low stage cancer group. Co-expressed genes participate in biological pathways associated with cell proliferation and inflammation, and anti-correlated genes are associated with mitochondrial genes and metabolism. Finally, *in vitro* experiments show decreased migration of cells knocked down for COL7A1, leading to the hypothesis that COL7A1 may have a functional role in the aggressiveness of ccRCC.

Overall, the incorporation of COL7A1 expression as a valid biomarker into increasingly personalized predictive tools may help to predict outcomes for patients with renal cell carcinoma. Of note, patients with intermediate stages, for whom the risk is difficult to evaluate, are well classified by COL7A1 expression.

## Figures and Tables

**Figure 1 cancers-15-02701-f001:**
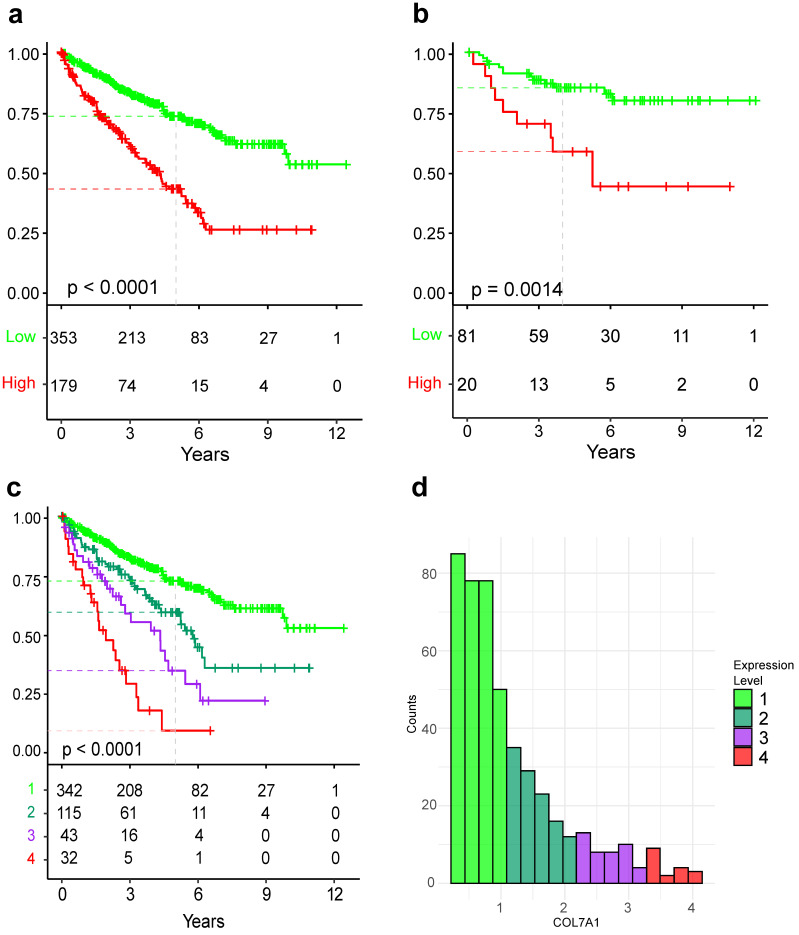
Kaplan–Meier curves showing differences in overall survival probability of patients that are grouped according to COL7A1 expression of the tumor. (**a**) Kaplan–Meier curve based on the TCGA:KIRC cohort, where patients are grouped into groups of low (green) and high (red) COL7A1 expression. (**b**) Same approach as in subfigure (**a**), applied to the E-MTAB-1980 cohort. (**c**) Kaplan–Meier curve based on the TCGA:KIRC cohort, where patients are grouped into four groups of increasing COL7A1 expression. (**d**) Histogram of COL7A1 expression in log2(TPM + 1), where colors correspond to groups of subfigure (**c**).

**Figure 2 cancers-15-02701-f002:**
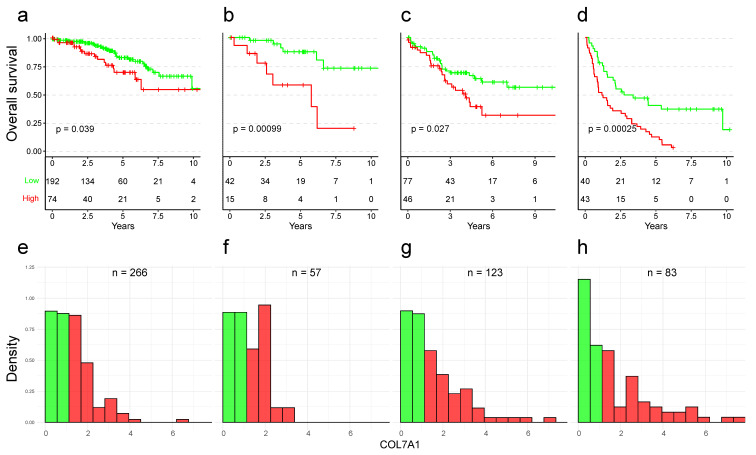
Kaplan–Meier curves and corresponding histograms showing differences in overall survival in the TCGA:KIRC cohort. Patients were split into four subsets according to their stage. Patients were divided into groups with low (green) and high (red) COL7A1 expression. (**a**,**e**) Stage I patients. (**b**,**f**) Stage II patients. (**c**,**g**) Stage III patients. (**d**,**h**) Stage IV patients. The expression threshold between the low and high expression groups is the same as in Figure 1a.

**Figure 3 cancers-15-02701-f003:**
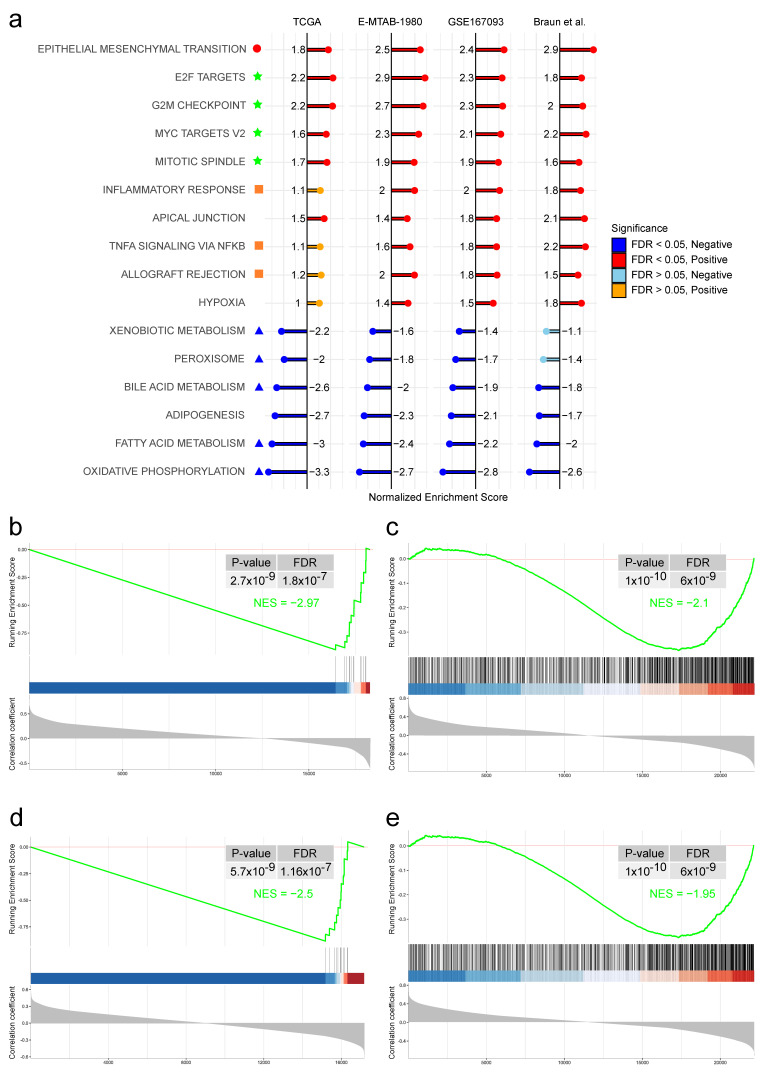
Results from GSEA analysis based on COL7A1 correlation, performed on four different datasets (TCGA:KIRC, E-MTAB-1980, GSE167093 and Braun *et al.*). (**a**) Most enriched HALLMARK pathways among all four datasets. Green stars highlight pathways related to cell proliferation, orange squares for inflammation/immune response, a red circle for the epithelial to mesenchymal transition pathway, and blue triangles for metabolic pathways; (**b**) GSEA plot of mitochondrial genes in the TCGA:KIRC cohort; (**c**) GSEA plot of GO:CC MITOCHONDRION pathway in the E-MTAB-1980 cohort; (**d**) Same as on subfigure (**b**), applied on the Braun *et al*. data; (**e**) Same as on subfigure (**c**), applied on the GSE167093 dataset.

**Figure 4 cancers-15-02701-f004:**
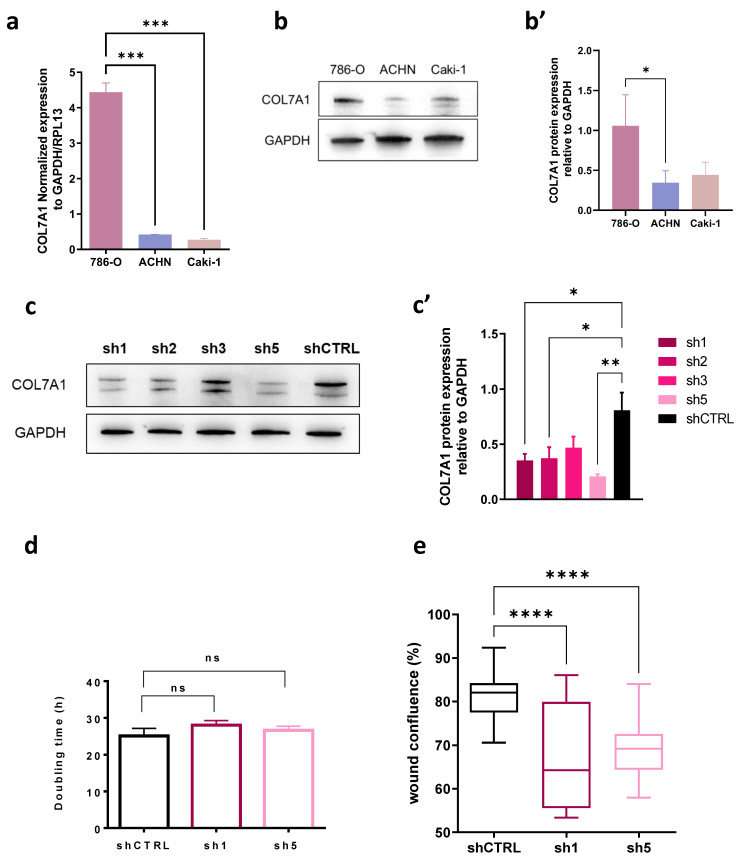
Influence of COL7A1 on proliferation and migration of 786-O ccRCC cells. (**a**): RT-qPCR showing the expression level of COL7A1 in three RCC cell lines: 786-O, ACHN, and Caki-1 (n = 6). (**b**): Western blot of COL7A1 protein level in the three cell lines and the respective quantification relative to GAPDH (**b′**) (n = 3). (**c**,**c′**): Western blot of shCOL7A1 786-O cell lines generated with different shRNAs and its quantification (n = 3). (**d**): Proliferation assay showing doubling time of 786-O shCTRL, shRNA1, and shRNA5 cell lines (n = 4). €: Wound healing assay of 786-O shCTRL, shRNA1, and shRNA5 cell lines, graph represents wound confluence (%) 10 h after the scratch (n = 24). Statistical analysis was made using the unpaired *t*-test, *p* < * 0.05, ** 0.01, *** 0.001, **** 0.0001, ns: not significant.

**Table 1 cancers-15-02701-t001:** Clinicopathological characteristics of patients with ccRCC in four different datasets. Presented are datasets that have publicly available clinicopathological data.

	TCGA:KIRC	E-MTAB-1980	GSE167093	Braun *et al.*	JAVELIN Renal 101
**Total samples**	532	101	604	225	726
**Median age at diagnosis** **years (range)**	61 (26–90)	64 (35–91)	62 (23–85)	62 (30–88)	61(27–88)
**Sex, N (%)**					
Female	187 (35.15)	24 (23.8)	247 (41)	61 (27.1)	178 (24.5)
Male	345 (64.85)	77 (76.2)	357 (59)	164 (72.9)	548(75.5)
**Tumor grade ^a^, N (%)**					-
I	14 (2.7)	13 (13.1)	100 (18.8)	-	-
II	228 (43.5)	59 (59.6)	304 (57)	-	-
III	206 (39.3)	22 (22.2)	105 (19.7)	-	-
IV	76 (14.5)	5 (5.1)	24 (4.5)	-	-
Missing	8	2	71	225	726
**Tumor stage ^a^, N (%)**					
I	266 (50.3)	66 (65.3)	306 (50.7)	0	-
II	57 (10.8)	20 (19.8)	98 (16.2)	0	-
III	123 (23.2)	3 (3)	138 (22.8)	0	-
IV	83 (15.7)	12 (11.9)	62 (10.3)	225 (100)	726 (100)
Missing	3	0	0	0	0
**EVENTS**	175 (32.9)	23 (22.78)	-	173 (76.9)	358 (49.3) ^b^
**5 year survival ** **probability % (95% CI)**	62.9 (58.2–68.1)	79.6 (71.7–88.3)	-	20.4 (15.5–26.9)	-

^a^ Samples with missing information were not included in the percentage calculation. ^b^ Event is defined as progression of disease.

**Table 2 cancers-15-02701-t002:** Results from a univariable Cox model performed on individual genes of the TCGA:KIRC cohort. The table shows the top 10 genes with the lowest *p*-value. COL7A1 is found at the top of the table. HR means Hazard Ratio.

Gene Symbol	*p*-Value	Coefficient	HR	Rank
COL7A1	5.1 × 10^−19^	0.44	1.56	1
CDCA3	6.2 × 10^−19^	0.74	2.09	2
CRB3	2.1 × 10^−18^	−0.50	0.60	3
SOWAHB	2.5 × 10^−18^	−0.52	0.60	4
SLC16A12	4.3 × 10^−17^	−0.28	0.75	5
SORBS2	1.9 × 10^−16^	−0.49	0.61	6
CYFIP2	2.3 × 10^−16^	−0.59	0.55	7
METTL7A	2.7 × 10^−16^	−0.55	0.57	8
TROAP	5.2 × 10^−16^	0.61	1.83	9
RGS17	1.1 × 10^−15^	0.76	2.14	10

**Table 3 cancers-15-02701-t003:** Results of univariable and multivariable Cox models learned on stage, grade, age of patients, and COL7A1 expression in the TCGA:KIRC cohort. The table also contains results from an ANOVA performed on the multivariable Cox model.

	Univariable Cox Model				Multivariable Cox Model ^b^			Anova	
Covariate	Coefficient (β)	HR [exp(β)]	*p*-Value	Score Test	Coefficient (β)	HR [exp(β)]	*p*-Value	Chisq	*p*-Value
STAGE	-	-	1.1 × 10^−26^	123.8	-	-	-	95.22	1.7 × 10^−20^
II	0.22	1.245	0.486	-	0.18	1.20	0.56		
III	0.91	2.48	1.2 × 10^−05^	-	0.55	1.73	0.013		
IV	1.84	6.32	1.4 × 10^−21^	-	1.44	4.23	1.8 × 10^−10^		
GRADE^a^	-	-	5.6 × 10^−18^	79.46	-	-	-	14.2	8.3 × 10^−4^
III	0.64	1.90	6.9 × 10^−4^	-	0.36	1.43	0.067		
IV	1.63	5.10	6.7 × 10^−16^	-	0.60	1.83	0.011		
AGE	0.035	1.03	2.5 × 10^−06^	22.51	0.036	1.036	1.4 × 10^−6^	22.67	1.9 × 10^−6^
COL7A1	0.44	1.56	5.8 × 10^−20^	83.67	0.32	1.37	6.9 × 10^−9^	28.45	9.6 × 10^−8^

^a^ Histological grades I and II are combined due to the low number of patients in grade I. ^b^ The Logrank test score for multivariable analysis is 215.3, and the likelihood ratio test is 160.5.

## Data Availability

Publicly available datasets were analyzed in this study. These data can be found here: https://www.cancer.gov/tcga; https://www.ebi.ac.uk/arrayexpress/experiments/E-MTAB-1980/; supplementary data of [29]; https://www.ncbi.nlm.nih.gov/geo/query/acc.cgi?acc=GSE167093; The R script to reproduce the results presented in this article is available at https://github.com/IRIG-BCI-IMAC/COL7A1-project (all accessed on 19 December 2022).

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
