# Peer review of "COL7A1 Expression Improves Prognosis Prediction for Patients with Clear Cell Renal Cell Carcinoma Atop of Stage"

_cancers, 2023, doi:10.3390/cancers15102701_

Round 1

Reviewer 1 Report (Previous Reviewer 1)

The Authors addressed an interesting topic reporting findings about screening for potential biomarker genes in ccRCC. They have shown that COL7A1 expression can stratify patients in terms of prognostic in independent cohorts, including recent ones dealing with up-to-date treatments. The paper needs revision before publication:

- I suggest shortening the introduction which remains well written.

-I appreciate the hints on ccRCC management especially the consideration for ablation approves which will soon acquire much space. For this purpose, please consider citing this paper on the comparison of ablation vs RAPN even in challenging indications (10.1089/end.2022.0478).

- I strongly suggest a makeover for materials and methods section and its synthesis to make the manuscript more readable. 

- When presenting Cox model and in general statistic analysis do not report the statistic theory when it is not the focus of the manuscript. Please remove considerations that I consider unnecessary.

-as your manuscript focuses on ccRCC, recent findings focused through an integrated approach using transcriptomics, metabolomics, and lipidomics on metabolic profiling of this cancer. I do believe it is worthily of interest and should be included in your manuscript.

-Check typos 

Author Response

The Authors addressed an interesting topic reporting findings about screening for potential biomarker genes in ccRCC. They have shown that COL7A1 expression can stratify patients in terms of prognostic in independent cohorts, including recent ones dealing with up-to-date treatments. The paper needs revision before publication:

We thank the reviewer for his general supportive comments. We have taken into account the comments below to improve our manuscript.

- I suggest shortening the introduction which remains well written.

We have removed unnecessary parts in the introduction, so that the text is more focused. In particular, we removed the sentences about discoloration of cancer cells and about type I collagen, which are not in direct links with our analysis.

-I appreciate the hints on ccRCC management especially the consideration for ablation approves which will soon acquire much space. For this purpose, please consider citing this paper on the comparison of ablation vs RAPN even in challenging indications (10.1089/end.2022.0478).

We have added in the previous round of revision the reference (DOI: 10.23736/S2724-6051.22.05092-3) above mentioned. We think that this new encouragement to add a paper from the same group is contradictory to the shortening of the introduction suggested previously.

- I strongly suggest a makeover for materials and methods section and its synthesis to make the manuscript more readable.

- When presenting Cox model and in general statistic analysis do not report the statistic theory when it is not the focus of the manuscript. Please remove considerations that I consider unnecessary.

We answer here to both related comments above. We thank the reviewer to encourage removing unnecessary digressions to enhance the clarity of the Material and Methods section. As tracked in red in the updated version, we have largely changed the ‘Material and Methods’ section. In particular, we have made a summary paragraph as suggested, and removed the part related to the theory of the Cox model.

-as your manuscript focuses on ccRCC, recent findings focused through an integrated approach using transcriptomics, metabolomics, and lipidomics on metabolic profiling of this cancer. I do believe it is worthily of interest and should be included in your manuscript.

We have added a reference to the use of integrated omics in the process of inferring causal molecular reasons for COL7A1 high expression.

-Check typos

We have re-read and corrected the whole manuscript.

Reviewer 2 Report (New Reviewer)

1) Authors should note that confluence is not a standard measure of cell proliferation, since it may be affected by morphological changes. Authors should add another measurement, such as cell counts at the beginning and end of the experiment, ki-67 expression, MTT or similar. 2) Authors should acknowledge that their in vitro work was done in 2D, which may be different from 3D results. This piece was missing from their discussion. 3) Authors should include another limitation in their discussion, which is that the in vitro experiments performed are not related to the changes predicted in their analysis. 4) This reviewer is curious about the prevalence of Col7 in biopsies. Can it be found? Has it been similarly related to overall survival changes? Is this protein potentially targetable?

Author Response

This paper was well design, is well written and methods are clearly explained. However, this reviewer was not able to find the main figures.

We thank the reviewer for his positive comments. We are sorry that the reviewer has not received the Figure files, nor our point-by-point response document to the previous reviewers. We have gathered all the comments in a single file to avoid similar issues. Please find below our answers to the specific points raised.

1) Authors should note that confluence is not a standard measure of cell proliferation, since it may be affected by morphological changes. Authors should add another measurement, such as cell counts at the beginning and end of the experiment, ki-67 expression, MTT or similar.

We are grateful to the reviewer for this suggestion, as we had not interpreted our result in the best manner. As you suggested, we directly measured cell proliferation, by counting the cells at t-0 (reference time) and 48 hours after. Indeed, cell proliferation was not affected by the knockdown of COL7A1: control cells have a doubling time of 25.1h, (95% confidence interval (CI) = 23.1-27.1) and knocked-down cells have a doubling time of 28.0h for sh1 and 26.6 for sh5 (95%CI 26.8-29.3 for sh1 and 95%CI 25.5-27.7), and knocked-down cells are bigger, at least at the beginning of the experiment. We have thus replaced Fig. 4d and changed the text accordingly.

2) Authors should acknowledge that their in vitro work was done in 2D, which may be different from 3D results. This piece was missing from their discussion.

We agree that the experiments are all performed in 2D models, which are not as relevant as 3D models. The core of the work is related to the prognosis potential in patients. The experiments performed led to raise the hypothesis that COL7A1 may have a causal role in the lower survival of patients, with a mechanism that remains to be elucidated. This is why we have not performed 3D experiments. We have added these points in the discussion.

3) Authors should include another limitation in their discussion, which is that the in vitro experiments performed are not related to the changes predicted in their analysis.

We have modified the “experimental validation” paragraph so that it better shows the motivations of performing the experiments. We have also added the limitation as requested.

4) This reviewer is curious about the prevalence of Col7 in biopsies. Can it be found? Has it been similarly related to overall survival changes? Is this protein potentially targetable?

We thank the reviewer for these interesting questions. While little is known related to COL7A1 in cancer, type I collagen is already a studied as a therapeutic target. We have now cited a review in the discussion session. COL7A1 tumor expression has been shown to be correlated with survival in lung squamous cell carcinoma, esophageal squamous cell carcinoma, gastric cancer and more recently pancreatic cancer. We have added the reference related to pancreatic cancer in the introduction.

Round 2

Reviewer 1 Report (Previous Reviewer 1)

In my opinion, the revised manuscript is now suitable for publication

Reviewer 2 Report (New Reviewer)

Thanks for addressing my comments appropriately.

This manuscript is a resubmission of an earlier submission. The following is a list of the peer review reports and author responses from that submission.

Round 1

Reviewer 1 Report

The authors should be congratulated for the good methodological approach provided and for the interesting topic. They performed (1) a screening for potential biomarker genes in ccRCC followed by an in-depth investigation of the top results, (2) performed in vitro experiments to investigate the impact of COL7A1 expression on the proliferation and migration of ccRCC cell lines, and (3) the possible biological implications of COL7A1 through the assessment of co-expressed genes. The manuscript is well-written and structured, however, a major revision is required.

- Introduction: "around 17% of newly diagnosed patients have developed distant metastasis". You already presented this data previously. Please eliminate repetition.

- When talking about novel therapeutical options compared to surgery, percutaneous ablation is needed to be included. For this scope please cite one or both among those multi-institutional reports (DOI: 10.23736/S2724-6051.22.05092-3;)

- When presenting results, please be more synthetic. Please leave to the discussion any comments and provide only aseptic results. Please translate all comments into the discussion.

- Considering that the metabolomic approach was used to study ccRCC biology, expand the discussion by including this aspect. Therefore, the clinical application of this field is metabolic biomarker detection. For the scope please include this up-to-date review of the topic

- Study limitations are needed before the conclusion.

- I strongly suggest filling a separate section for the conclusion.

Reviewer 2 Report

The authors describe COL7A1, matrix protein, expression can be a new prognostic biomarker in clear cell renal cell carcinoma, using TCGA and KIRC database. They also report that COL7A1 knockdown reduced migration and proliferation activities by in vitro study.

 The submitted cohort had relatively short-term follow-up data. Therefore, some prognostic analyses could not prove the authors’ conclusion. For example, more than half of the patients seemed to be censored data in several groups. Current medications, such as tyrosine kinase inhibitors and immune checkpoint inhibitors, can prolong patients’ outcomes. Looking at the authors’ data, their prognosis is far worse compared to those of the current status. Unfortunately, the follow-up data of the TCGA and KIRC databases were established before the current drug sequence strategy. The authors need to create their original cohort data, which reflects the current practice.

Table 3.:

Some analytical data are doubtful. For example, the p-value of age was 1.4X10-6, although its hazard ratio was 1.036. The grading hazard ratio was too small compared to published data. The authors need to consult the data analysis a biostatistician.